# From Snake Venoms to Therapeutics: A Focus on Natriuretic Peptides

**DOI:** 10.3390/ph15091153

**Published:** 2022-09-16

**Authors:** Wei Fong Ang, Cho Yeow Koh, R. Manjunatha Kini

**Affiliations:** 1Department of Biological Sciences, Faculty of Science, National University of Singapore, Singapore 117558, Singapore; 2NUS Graduate School of Integrative Sciences and Engineering, National University of Singapore, Singapore 119077, Singapore; 3Department of Medicine, Yong Loo Lin School of Medicine, National University of Singapore, Singapore 117559, Singapore; 4Department of Pharmacology, Yong Loo Lin School of Medicine, National University of Singapore, Singapore 117600, Singapore; 5Department of Biochemistry and Molecular Biology, Virginia Commonwealth University, Richmond, VA 23298-0614, USA

**Keywords:** natriuretic peptide, natriuretic peptide receptor, venom peptides, therapeutics

## Abstract

Snake venom is a cocktail of multifunctional biomolecules that has evolved with the purpose of capturing prey and for defense. These biomolecules are classified into different classes based on their functions. They include three-finger toxins, natriuretic peptides, phospholipases and metalloproteinases. The focus for this review is on the natriuretic peptide (NP), which is an active component that can be isolated from the venoms of vipers and mambas. In these venoms, NPs contribute to the lowering of blood pressure, causing a rapid loss of consciousness in the prey such that its mobility is reduced, paralyzing the prey, and often death follows. Over the past 30 years since the discovery of the first NP in the venom of the green mamba, venom NPs have shown potential in the development of drug therapy for heart failure. Venom NPs have long half-lives, different pharmacological profiles, and may also possess different functions in comparison to the mammalian NPs. Understanding their mechanisms of action provides the strategies needed to develop new NPs for treatment of heart failure. This review summarizes the venom NPs that have been identified over the years and how they can be useful in drug development.

## 1. Introduction

Venoms are a cocktail of bioactive molecules, and usually contain proteins and peptides as their main component [1,2]. They are used as a biological/chemical weapon in animals in addition to physical force for predatory purposes and sometimes as a defense mechanism when they are in danger. Animals that have venom include snakes, scorpions, spiders, cone snails and others. Depending on the active components in the venom, different venoms have their own unique effects. Within the snake venoms, their bioactivities come in the form of neurotoxicity, hemotoxicity, coagulopathy and cytotoxicity, or even a combination of different bioactivities [1]. These venom compositions vary between species [3] and also within species and can be influenced by factors such as age [4], gender [5], type of prey [6] and other environmental factors [1,7]. Hence, snake venoms are known to be a cocktail that is complex and multifunctional. Despite the association of snake venom with pain and death, translational research has made good use of these venoms for basic science as well as the development of therapeutics inspired by snake venom toxins [8,9]. This review discusses the current knowledge about the natriuretic peptide, a vasorelaxant that can be found in some snake venom, and how this knowledge can be used to develop therapeutic treatments in areas such as hypertension and heart failure.

## 2. Natriuretic Peptides (NPs): General Considerations

Natriuretic peptides (NPs) are probably the first biological molecules to demonstrate the link between two distinctly far organs: the heart and the kidney. In 1981, it was found that injecting atrial homogenates but not the ventricle homogenates into rats intravenously reduced their blood pressure and increased the excretion of sodium and water [10]. This was later identified and named as atrial natriuretic peptide (ANP). Subsequently, the discovery of ANP led to blooming research on NPs from the 1980s up to now. Three variants of NPs have been found in the mammals, atrial natriuretic peptide (ANP), b-type natriuretic peptide (BNP) and c-type natriuretic peptide (CNP) [11,12,13]. All three mammalian NPs possess the conserved 17-residue ring structure (CFGXXXDRIXXXXGLGC, where X represents any amino acid residues) held together by a single disulfide bond (Figure 1) [14]. This 17-residue ring is essential for the binding to their receptors and the variation in the amino acid residues within this ring structure confer their specificity towards the receptors [15]. These NPs are also highly variable in their N-terminal and C-terminal sequences, which differentiate their functions from one another [15].

All three mammalian NPs are synthesized as a preprohormone, and upon cleaving of the signal peptide, prohormone is formed (Figure 1) [11,16,17]. Both proANP and proBNP are synthesized in the heart and are stored in the atrial granules [18,19]. CNP is synthesized and stored in endothelial cells [13]. In the presence of a stimulus such as a stretch in the arterial or ventricular walls caused by an increasing intra-cardiac filling pressure, these prohormones are cleaved into their mature form and secreted (28-residue ANP, 32-residue BNP, 22-residue CNP and 53-residue CNP) (Figure 1) [11,13]. The release of these prohormones can also be stimulated by the actions of other hormones such as endothelin, angiotensin and vasopressin [11]. Cleaving of proBNP into mature BNP also produces another product, NT-proBNP, which serves as a useful prognostic biomarker for heart failure (HF) [13,20]. Cardiac ANP and BNP circulate in the bloodstream to reach their respective receptor in the effector organs to elicit their functions in an endocrine manner, while CNP is produced locally and participates in paracrine action of other vasorelaxant biomolecules [11,13]. Their main functions are to induce vasodilation, natriuresis and diuresis to bring about a reduction in the blood pressure. Among the three mammalian NPs, ANP exhibits the strongest natriuretic bioactivities, while CNP plays a lesser role in natriuresis. Interestingly, BNP and CNP have other bioactivities. Knocking out BNP in mice leads to the development of fibrosis in heart ventricles, suggesting that BNP has anti-fibrotic bioactivity [13,21]. CNP, which is highly expressed in the brain and chondrocytes, is involved in the chondrocyte differentiation process, cardiac remodeling and bone formation [13,22]. It is found that administrating CNP into rats with myocardial infarction reduces cardiac remodeling and improves cardiac function, probably due to its anti-fibrotic and anti-hypertrophic functions [13].

All three mammalian NPs elicit their functions by binding to their respective receptors. There are three natriuretic peptide receptors (NPRs) that are known in mammals: NPR-A, NPR-B and NPR-C [23]. Both NPR-A and NPR-B belong to the membrane-bound guanylyl cyclase family where they consist of a catalytic guanylyl cyclase (GC) domain in their intracellular region (Figure 2) [23,24]. In contrast, NPR-C lacks this GC domain and is believed to function as the clearance receptor for all three NPs [23,24]. Both ANP and BNP bind to NPR-A, with ANP having a stronger binding affinity, while CNP selectively binds to NPR-B [25]. All three mammalian NPs bind with similar affinity to NPR-C [25,26]. Upon binding of NPs to NPR-C, the peptides becomes internalized and degraded by lysosomes [27]. Other than the clearance receptor, NPs are also subjected to degradation by neutral endopeptidases (NEP) [28]. As such, the half-lives of NPs differ, with ANP having the shortest half-life of 2 min [29], followed by CNP (2.6 min) [30], and BNP with the longest half-life of 20 min (Figure 2) [31].

## 3. Snake Venom Natriuretic Peptides (NPs)

NPs are found mainly in the venoms of two families of venomous snake: *Elapidae* (elapids) and *Viperidae* (vipers) [32]. The toxicity of snake venom NPs is due to a rapid loss of consciousness through vasorelaxation and reduction in the contractility of the myocardium [32]. Snake venom NPs share the same 17-residue ring structure as other mammalian NPs and differ in the length and sequence of the N-terminal and C-terminal tails.

### 3.1. Dendroaspis NP (DNP) and Chimeric CD-NP 

DNP was the first NP isolated from snake venom. This 38-residue peptide originates from the green mamba snake *Dendroaspis angusticeps* [33]. Singh et al. evaluated the vasodilatory function of DNP in cultured aortic endothelial cells and myocytes. They found that DNP has comparable potency with ANP but not CNP [34]. This suggested that DNP selectively binds to NPR-A and not NPR-B in smooth muscle cells to produce cGMP, which induces vasodilation [33]. It also completely reversed the endothelin-1 (ET-1)-induced vasoconstriction in the mammary artery in the presence or absence of endothelium [34]. DNP also exhibits renal function (natriuresis and diuresis) similar to ANP and BNP [35]. However, DNP has a much longer C-terminal tail (15 residues) than ANP (5 residues), resulting in a 30-fold lower affinity towards NPR-C and a 60-fold higher resistance towards NEP degradation [33]. Thus, DNP has a longer half-life of 120 min as opposed to ANP (2–4 min). Interestingly, Schirger et al. detected the presence of a DNP-like peptide (DNP-L1) in human plasma and atrial myocardium. This peptide was found to be elevated in congestive heart failure (CHF) patients [36]. DNP-L1 displayed similar functions as ANP and it caused stronger vasodilation in arteries than in veins [37]. DNP-L1 is now thought to be the fourth member of the mammalian NPs. Subsequently, another group found that intravenous injection of synthetic DNP into CHF patients caused a significant reduction in the cardiac filling pressure, which is beneficial for the reduction of congestion in the heart [38]. These data suggested DNP has therapeutic potential for decompensated CHF.

This C-terminal tail of DNP was later conjugated with a mammalian CNP-ring to create a chimeric NP: CD-NP. CD-NP, also named as Cenderitide, was designed such that it can activate both NPR-A and NPR-B, retaining natriuretic, diuretic, anti-fibrotic, anti-proliferative and vascular regenerative effects of their respective parent peptides with an overall reduction in hypotensive side effects, compared to synthetic ANP and BNP therapy [39]. The addition of the long C-terminal tail of DNP to the CNP-ring extended the half-life of CD-NP by making CD-NP more resistant to NEP degradation compared to other NPs. This C-terminal extension also interacts with NPR-A to induce ANP functions such as vasodilation, diuresis and natriuresis (Figure 3A) [40]. On the other hand, the CNP-ring binds and activates NPR-B to induce vasodilation and anti-proliferative function (Figure 3A) [40]. Essentially, CD-NP can be considered a CNP analogue given that the CNP-ring is retained, and most of its function comes from the binding of this ring structure to the receptors. Hence, while CNP plays a limited role in renal function (natriuresis and diuresis), the addition of the C-terminal tail of DNP allows CD-NP to function like a CNP with some renal function. This could be the reason why CD-NP has fewer hypotensive side effects as compared to the synthetic ANP and BNP. Clinical trials show that CD-NP is safe and tolerable in patients with stable chronic HF [39,41]. Thus, this indicates Cenderitide is a promising drug candidate for HF treatment.

### 3.2. Krait NP (KNP), DGD-ANP and DRD-ANP

With the discovery of DNP, various venom NPs were subsequently isolated and characterized. One example is KNP, which was identified from the venom of the red-headed krait snake (*Bungarus flaviceps*) using transcriptomic studies [42]. KNP consists of a 17-residue ring structure (named K-ring) similar to the mammalian NPs. It has a longer C-terminal tail made up of 38 amino acid residues, and this tail has the propensity to form an 𝛼-helix (last 21 residues, named Helix) [43]. A structural and functional study analyzing the different segments of KNP demonstrated that KNP contains two functional segments: the K-ring and the Helix (Figure 3B) [43]. The K-ring binds to NPR-A like a typical NP and activates the cGMP-mediated pathway. In contrast, the Helix binds to an unknown receptor to induce nitric-oxide-dependent vasodilation [43]. This indicated that KNP functions differently compared to a typical NP. Infusion of KNP in anaesthetized rats resulted in a prolonged and sustained drop in blood pressure. KNP has a lower potency (10-fold) in activating NPR-A compared to ANP [43]. KNP induced endothelium-dependent vasodilation with absolutely undetectable renal effects even at 100 times higher concentration than ANP. Thus, this difference in the hemodynamic effect of the K-ring shows the possibility of delineating the vasodilatory properties and renal properties (natriuresis and diuresis) of NPs through a comparative study of structure–function relationships.

Using various chemically synthesized K-ring mutants and ANP mutants, three amino acid residues at positions 3, 4 and 14 of the ANP ring as well as the C-terminal tail of ANP (sequence: NSFRY) were identified to be responsible for demarcating the vasodilatory properties from the renal properties of NPs [44]. Comparing the ANP-ring to K-ring, glycine residues at position 3, 4 and 14 in the ANP-ring are substituted with aspartate, arginine and aspartate in the K-ring, respectively. Glycine at position 3 and 14 is conserved in all NPs except the K-ring. It was found that either G3D or G14D substitution lowered the potency in activating NPR-A. The K-ring with both D3G and D14G substitution became a mutant that gained renal effects [44]. This suggested that amino acid residues at position 3 and 14 determine whether the NPs will have a renal function in addition to their vasodilatory function. Amino acid residues at position 4 are different among the NPs with glycine in ANP, arginine in both BNP and KNP, leucine in CNP and histidine in DNP. This particular residue at position 4 helps to enhance the potency in activating NPR-A if the residues at position 3 and 14 are aspartate [44]. Finally, the C-terminal tail of ANP (NSFRY) appears to act as a force diuretic switch, where the presence of this C-terminal tail caused otherwise exclusively vasoactive peptides to gain renal effects, and this change in the function is independent of the presence of aspartate at either position 3 or 14 [44].

Using this knowledge, two ANP analogues were synthesized using solid-phase peptide synthesis, with one having exclusively renal function (DGD-ANP) and the other exclusively vascular function (DRD-ANP) [44,45]. The ANP analogue, DGD-ANP, was designed by substituting the glycine at position 3 and 14 with aspartate in ANP (Figure 3C). This analogue displayed an increase in urine output but did not reduce the mean arterial blood pressure (MAP) in anaesthetized rats [44]. Hence, DGD-ANP exhibits only renal function with no vascular function. Similar observations were also obtained when DGD-ANP was infused into healthy sheep and heart failure (HF)-induced sheep [45]. DGD-ANP exhibited only diuresis/natriuresis similar to ANP but no hypotensive activity in both healthy and HF sheep even at a concentration that was 10-fold higher. Interestingly, DGD-ANP did not affect the level of potassium excreted, suggesting that this analogue also served as a potassium-sparing agent [45]. This unique renal active DGD-ANP provides a safer and effective treatment for acute decompensated heart failure (ADHF) patients that suffer from congestion in the heart with no hypoperfusion. This is because DGD-ANP can induce diuresis and natriuresis, which reduced the congestion in the heart without lowering the MAP. The other ANP analogue, DRD-ANP, was created by replacing glycine at position 3, 4 and 14 of the ring structure in ANP with the residues found in the K-ring (aspartate–arginine–aspartate, respectively, Figure 3C). Infusing a low dose (0.2 nmol/kg/min) of DRD-ANP into anaesthetized rats caused a drop in MAP with no increase in urine output [44]. However, at a 10-fold higher dose (2 nmol/kg/min), there is both a reduction in MAP and increase in urine output [44]. Hence, DRD-ANP is an ANP analogue with exclusively vasodilatory properties when given at a low dose. In healthy and HF sheep, infusing DRD-ANP lowers the MAP but has a negligible increase in water and sodium excretion [45]. This makes vasoactive DRD-ANP ideal for ADHF patients that suffer from hypoperfusion with no congestion in the heart. Its ability to reduce MAP without increasing HR and without reducing the overall body fluid volume makes this drug candidate more tolerable and safer, with fewer side effects [45].

### 3.3. Other Snake NPs

Other than DNP and KNP as described above, many other snake venom NPs have also been reported and have potential for therapeutic development. In this section, we highlight some of the key findings in venom NPs isolated from snakes. Venom NPs isolated from the inland taipan snake (*Oxyranus microlepidotus*) consist of three isoforms, namely TNP-a, TNP-b and TNP-c [46]. All three NPs have a long C-terminal tail similar to DNP but only TNP-c shows vasodilatory properties. This difference in the activities of the three variants is due to the presence of a proline residue replacing the conserved hydrophobic residue at the third position in the ring structure of TNP-a and TNP-b [46]. More NP isoforms were also identified from other Australian elapids, namely eastern brown snake (*Pseudonaja textilis*) and king brown or mulga snake (*Pseudechis australis*), using NP-specific DNA amplification [47]. Among these venom NPs, only Pt-NP-a and Pt-NP-c were recombinantly expressed, purified and characterized. Both Pt-NP-a and Pt-NP-c inhibited angiotensin-converting enzyme in a dose-dependent manner, giving rise to the vasoactive function of the venom [47]. Out of these two NPs, only Pt-NP-a produces cGMP, indicating that it has normal natriuretic activity. There are also venom NPs that are glycosylated. For example, TcNP-a, isolated from the venom of the rough scaled snake (*Tropidechis carinatus*), undergoes O-glycosylation and has a galactosyl-β(1-3)-N-acetylgalactosamine modification located at a threonine residue in the C-terminal tail. Both glycosylated and non-glycosylated TcNP-a can activate both NPR-A and NPR-B. The activation of NPR-B is 3-fold higher than that of NPR-A.

Some of these venom NPs exhibit functions other than typical renal- and vessel-related activities. For instance, two NP-like peptides (lebetins 1 and 2) isolated from the venom of the blunt-nosed viper (*Macrovipera lebetina*) have anti-platelet properties, and these are controlled by their N-terminal sequence [48]. A similar N-terminal sequence found in another NP-like peptide in the Persian horned viper (*Pseudocerastes persicus*) also exhibits anti-platelet properties [49]. Recently, lebetin-2 showed acute, strong and prolonged cardioprotective effects in acute myocardial ischemia-reperfusion (IR) [50]. In IR mice, lebetin-2 reduces inflammatory response and fibrosis while recruiting more anti-inflammatory M2-like macrophages [50]. Table 1 summarizes all the venom NPs discussed in this review.

## 4. Conclusions

Identifying and characterizing snake venom NPs provides a platform for understanding the basic molecular and mechanistic characteristics of NPs and their interactions with NPRs, from which the knowledge can be applied and used to design drug therapy, as seen in CD-NP, DGD-ANP and DRD-ANP.

## Figures and Tables

**Figure 1 pharmaceuticals-15-01153-f001:**
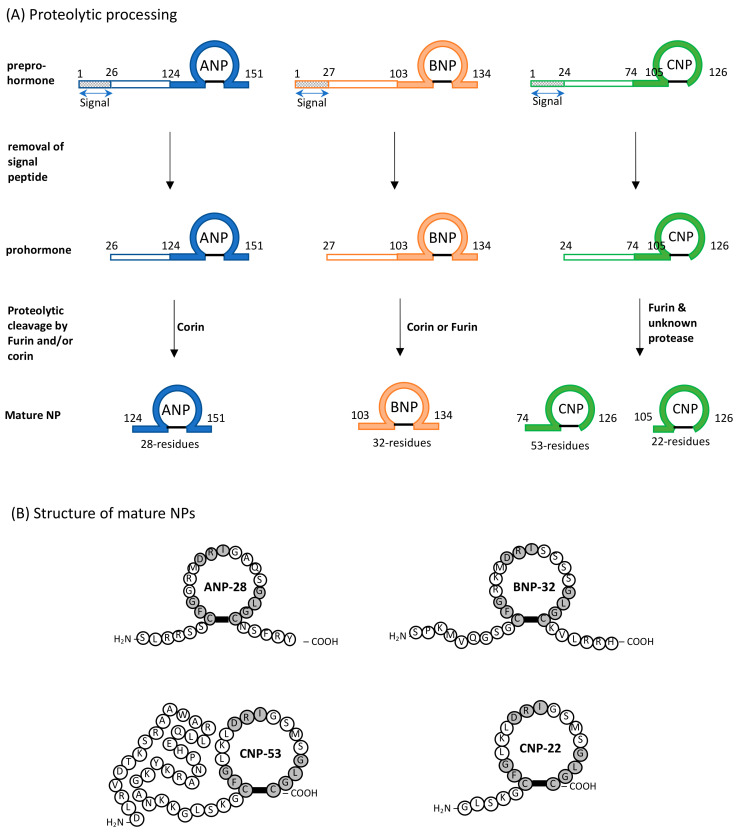
Proteolytic processing and mature structure of ANP, BNP and CNP. (**A**) All three mammalian NPs are expressed as a preprohormone that is cleaved to give prohormone with the removal of the signal peptide. This prohormone is further cleaved to give the mature form (ANP-28, BNP-32, CNP-22 and CNP-53). The cleaving from prohormone to the mature state is governed by endogenous proteases (furin/corin). (**B**) All three NPs share a conserved 17-residue ring with a single disulfide bond (indicated by a dark line) and vary in the N- and C-termini. The conserved residues are highlighted in gray. Data extracted from [11].

**Figure 2 pharmaceuticals-15-01153-f002:**
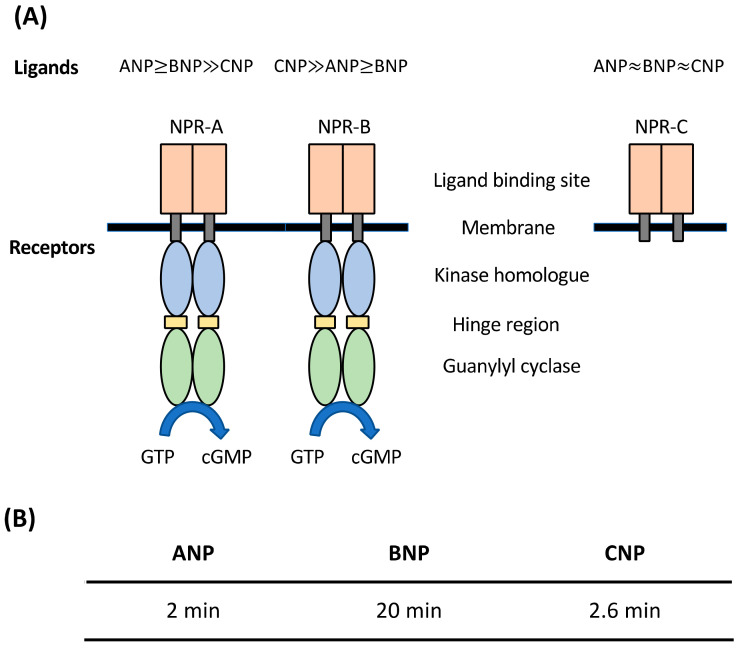
Schematic illustrations of NPRs and half-lives of NPs. (**A**) Both NPR-A and NPR-B belong to guanylyl cyclase receptor family but not NPR-C. Both ANP and BNP are selective towards NPR-A while CNP is more selective towards NPR-B. All three NPs have equal binding affinity to NPR-C. (**B**) ANP has the shortest half-life, followed by CNP and then BNP.

**Figure 3 pharmaceuticals-15-01153-f003:**
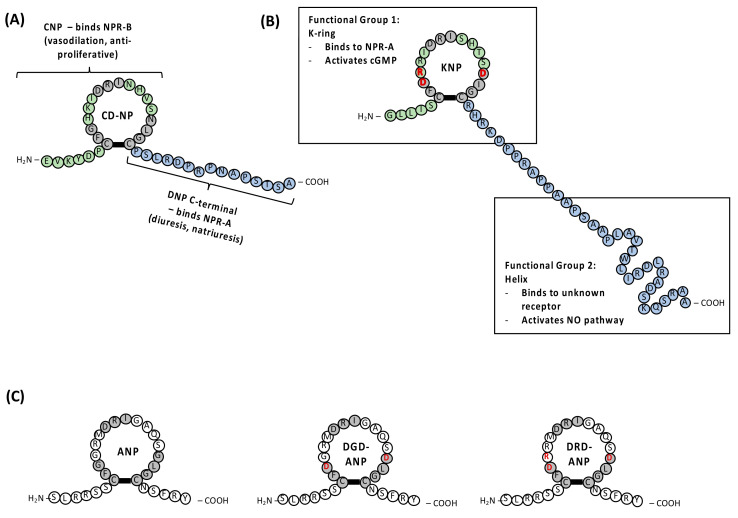
Structure of CD-NP, KNP, ANP and ANP analogues. (**A**) CD-NP made up of CNP with DNP C-terminal tail. It binds to NPR-B and NPR-A. (**B**) KNP has two functional groups, a K-ring that binds to NPR-A and a Helix tail that activates the NO-dependent pathway. (**C**) Structure of DGD-ANP and DRD-ANP compared to parent ANP. Changes in amino acid residues are in red font.

**Table 1 pharmaceuticals-15-01153-t001:** A summary of venom NPs and their functions as discussed in this review.

NPs	Snakes	Functions	Applications
DNP	Green mamba (*Dendroaspis angusticeps*)	Vasodilation, natriuresis, diuresis	C-terminal tail creates CD-NP with CNP
KNP	Red-headed krait (*Bungarus flaviceps*)	Vasodilation	Create DGD-ANP and DRD-ANP
TNP-a, TNP-b and TNP-c	Taipan snake (*Oxyranus microlepidotus*)	Vasodilation (TNP-c)	-
Pt-NP-a, Pt-NP-c	Eastern brown snake (*Pseudonaja textilis*) King brown snake (*Pseudechis australis*)	Inhibit ACE ^1^	-
TcNP-a	Rough scaled snake (*Tropidechis carinatus*)	Activate both NPR-A and NPR-B	-
Lebetin 1 and 2	Blunt-nosed viper (*Macrovipera lebetina*)	Anti-platelet, reduce inflammation	-

^1^ ACE: angiotensin-converting enzyme.

## Data Availability

Not applicable.

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
