# Peer review of "From Snake Venoms to Therapeutics: A Focus on Natriuretic Peptides"

_pharmaceuticals, 2022, doi:10.3390/ph15091153_

Round 1

Reviewer 1 Report

This is a comprehensive, timely and well-written review on mammalian and snake venom natriuretic peptides and their three receptors with emphasis on therapeutic potential in the development of new drugs in the treatment of heart failure. 

Only minor issues deserve authors' attention:

Headings of paragraphs 2 and 3 are similar ''Natriuretic Peptides (NPs)''. Maybe identify paragraph 2 as Mammalian Natriuretic Peptides (NPs) and paragraph 3 as Snake Venom Natriuretic Peptides (NPs). 

Line 230: Fry et al., 2005 should be replaced by number 45.

Line 272: reference 3 is missing and embedded with reference 2.

Author Response

Reviewer 1

This is a comprehensive, timely and well-written review on mammalian and snake venom natriuretic peptides and their three receptors with emphasis on therapeutic potential in the development of new drugs in the treatment of heart failure. 

Only minor issues deserve authors' attention:

Headings of paragraphs 2 and 3 are similar ''Natriuretic Peptides (NPs)''. Maybe identify paragraph 2 as Mammalian Natriuretic Peptides (NPs) and paragraph 3 as Snake Venom Natriuretic Peptides (NPs). 

Thank you for the suggestion. We have changed the title of Section 2 to “Natriuretic peptides (NPs): General considerations” and the title of Section 3 to “Snake venom natriuretic peptides (NPs)”.

Line 230: Fry et al., 2005 should be replaced by number 45.

Line 272: reference 3 is missing and embedded with reference 2.

Apologies for the oversights, we have fixed the issues with references.

Reviewer 2 Report

Dear authors,

I recommend that a minor revision of the manuscript is warranted. I explain my concerns in more detail below and I would ask you to consider each of my comments in your next response.

Minor comments:

·         At correspondence line you should indicate the email addresses of both corresponding authors marked with * in the author list.

Line 21-22  - ” that mobility is no longer an option” –not scientifically described may be mobility is reduced .

Subchapter 2 and 3 have the same title, one should be changed!

Line 110/111 – the two families of venomous snake should be italic, they seems being from Latin (ending in –ae);

Line 114/115 – was mentioned previously, this phrase is not necessary

Line 179 and 196 - for the peptides described herein  such as various K-ring mutants and ANP mutants/ two ANP analogues I consider relevant to mention which method vas uses for synthesis and usually “created” is not the scientific word for synthesizing molecules.

Author Response

Reviewer 2

Dear authors,

I recommend that a minor revision of the manuscript is warranted. I explain my concerns in more detail below and I would ask you to consider each of my comments in your next response.

Minor comments:

At correspondence line you should indicate the email addresses of both corresponding authors marked with * in the author list.

Noted. We have added the “*” accordingly.

Line 21-22  - ” that mobility is no longer an option” –not scientifically described may be mobility is reduced .

We have modified this sentence to “…causing a rapid loss of consciousness in the prey such that its mobility is reduced, paralyzing the prey and often death follows.”

Subchapter 2 and 3 have the same title, one should be changed!

Thank you for the suggestion. We have changed the title of Section 2 to “Natriuretic peptides (NPs): General considerations” and the title of Section 3 to “Snake venom natriuretic peptides (NPs)”.

Line 110/111 – the two families of venomous snake should be italic, they seems being from Latin (ending in –ae);

Elapidae” and “Viperidae” are now italicised.

Line 114/115 – was mentioned previously, this phrase is not necessary

We have removed the sentence.

Line 179 and 196 - for the peptides described herein  such as various K-ring mutants and ANP mutants/ two ANP analogues I consider relevant to mention which method vas uses for synthesis and usually “created” is not the scientific word for synthesizing molecules.

We have changed the sentences as followed: “Using various chemically synthesized K-ring mutants and ANP mutants, three amino acid residues at positions 3, 4, and 14 of the ANP ring as well as the C-terminal tail of ANP…” (Line 179) and “Using this knowledge, two ANP analogues were synthesized using solid-phase peptide synthesis with one having exclusively renal function…” (Line 196).

Reviewer 3 Report

To Authors

General Considerations

The aim of this review article is to suggest that NPs are active components that can be isolated from the venoms of vipers and mambas. Therefore, the cNPs can be useful in drug development of drug therapy for heart failure.

This review is concise and clear. The data reported are interesting and in part original. I have only some minor points to address to Authors with the aim to further improve the scientific message of this interesting review.

 Specific Points

1.     The titles of the paragraphs 2 (line 49 page 2) and 3 (line 109, page 4) are the same (Natriuretic Peptides, NPs). I would like to suggest for the title of paragraph 2: Natriuretic Peptides: General Considerations, and for the paragraph 3: Natriuretic Peptides in the snake venom.

2.     The references (11 and 21) concerning the pathophysiological characteristics of CNP should be updated. For a more recent review I would like to suggest: Del Ry S. et al. Pharmacol Res 2013;76:190-198.

Author Response

Reviewer 3

General Considerations

The aim of this review article is to suggest that NPs are active components that can be isolated from the venoms of vipers and mambas. Therefore, the cNPs can be useful in drug development of drug therapy for heart failure.

This review is concise and clear. The data reported are interesting and in part original. I have only some minor points to address to Authors with the aim to further improve the scientific message of this interesting review.

 Specific Points

  1. The titles of the paragraphs 2 (line 49 page 2) and 3 (line 109, page 4) are the same (Natriuretic Peptides, NPs). I would like to suggest for the title of paragraph 2: Natriuretic Peptides: General Considerations, and for the paragraph 3: Natriuretic Peptides in the snake venom.

Thank you for the suggestion. We have changed the title of Section 2 to “Natriuretic peptides (NPs): General considerations” and the title of Section 3 to “Snake venom natriuretic peptides (NPs)”.

  1. The references (11 and 21) concerning the pathophysiological characteristics of CNP should be updated. For a more recent review I would like to suggest: Del Ry S. et al. Pharmacol Res 2013;76:190-198.

We have updated the references.